# Natural Killer Cell-Based Immunotherapy against Glioblastoma

**DOI:** 10.3390/ijms24032111

**Published:** 2023-01-20

**Authors:** Takayuki Morimoto, Tsutomu Nakazawa, Ryosuke Maeoka, Ichiro Nakagawa, Takahiro Tsujimura, Ryosuke Matsuda

**Affiliations:** 1Department of Neurosurgery, Nara Medical University, Kashihara 634-8521, Japan; 2Department of Neurosurgery, Nara City Hospital, Nara 630-8305, Japan; 3Grandsoul Research Institute for Immunology, Inc., Uda 633-2221, Japan; 4Clinic Grandsoul Nara, Uda 633-2221, Japan

**Keywords:** glioblastoma, NK cell, immunotherapy, immunosuppression, tumor microenvironment

## Abstract

Glioblastoma (GBM) is the most aggressive and malignant primary brain tumor in adults. Despite multimodality treatment involving surgical resection, radiation therapy, chemotherapy, and tumor-treating fields, the median overall survival (OS) after diagnosis is approximately 2 years and the 5-year OS is poor. Considering the poor prognosis, novel treatment strategies are needed, such as immunotherapies, which include chimeric antigen receptor T-cell therapy, immune checkpoint inhibitors, vaccine therapy, and oncolytic virus therapy. However, these therapies have not achieved satisfactory outcomes. One reason for this is that these therapies are mainly based on activating T cells and controlling GBM progression. Natural killer (NK) cell-based immunotherapy involves the new feature of recognizing GBM via differing mechanisms from that of T cell-based immunotherapy. In this review, we focused on NK cell-based immunotherapy as a novel GBM treatment strategy.

## 1. Introduction

Glioblastoma (GBM) is the most common and aggressive primary brain tumor. The annual incidence of GBM is 3.19 per 100,000 [1] and it is classified as grade IV by the World Health Organization [2]. For several decades, the standard GBM therapy has consisted of maximum safety resection, adjuvant radiotherapy, and chemotherapy with temozolomide, termed the Stupp regimen. Despite the multidisciplinary therapy, the median overall survival (mOS) is only 15–17 months and the 5-year overall relative survival is only 5.8% [3,4]. Several novel strategies were investigated, where the addition of tumor-treating fields to standard treatment statistically significantly improved progression-free survival and OS [5]. A recent phase 2 trial of intratumoral oncolytic herpes virus G47∆ for residual or recurrent GBM demonstrated survival benefits and a good safety profile, which led to the approval of G47∆, albeit with conditions under the early approval system of Japanese-specific law, as the first oncolytic virus product in Japan [6]. Figure 1 summarizes the multimodality treatment against GBM. However, given the poor prognosis of patients with GBM, further novel approaches are needed for GBM treatment.

Almost every immunotherapy mainly aims to generate a tumor-specific immune response to selectively eliminate tumor cells as a result of T cell activation. The representative immunotherapies are checkpoint inhibitor and chimeric antigen receptor (CAR) T-cell therapies, which have been used in other solid tumors and hematologic malignancies [7,8,9]. Contrastingly, several GBM immunotherapies have long been investigated, but few attractive strategies have been reported.

Compared to T cell-based therapy, natural killer (NK) cell-based therapy approaches tumors from new aspects. First, NK cells recognize the tumor, which consists of heterogeneous cells, via multiple activating and inhibitory receptors despite the diminished or absent expression of major histocompatibility complex class I (MHC-I) molecules [10]. Second, NK cells are important for recruiting conventional type 1 dendritic cells (cDC1s) and subsequently CD8^+^ T cells [11,12]. These functions promote cancer immunity cycle activation, which has the advantage of overcoming the immunosuppressive GBM tumor microenvironment (TME) [13,14]. In this review, we summarize the characteristics of GBM, the GBM TME, and GBM immunotherapy, mainly focusing on the prospective aspect of NK cell-based immunotherapy.

## 2. Glioblastoma: Molecular Features and the TME

GBM is histologically characterized by microvascular proliferation or necrosis in wild-type IDH [2]. GBM is molecularly characterized by *CDKN2A*/*B* homozygous deletion in mutant *IDH* and *TERT* promoter mutation, *EGFR* gene amplification or +7/−10 chromosome copy number changes in wild-type *IDH* [2]. The GBM TME consists of a small population of GBM stem cells (GSCs), resistant to chemotherapy and radiotherapy [15,16,17]. GSCs are maintained in hypoxic and periarteriolar GSC niches [18,19] and are more abundant in more aggressive high-grade tumors with worse prognosis [20]. Furthermore, GBM exhibits intratumoral heterogeneity, where one patient with GBM can demonstrate hypoxia and stem cell, resistance, transformed neuronal, proliferative, and mutation regions [21]. Intratumoral heterogeneity in recurrent GBM after adjuvant therapy demonstrates new, different heterogeneity from that in primary GBM [22]. These cytologic features render GBM treatment-resistant, which is one reason that single-molecular target therapies, such as molecular targeted drugs and CAR-T therapy, cannot yield sufficiently favorable outcomes.

The central nervous system (CNS) is considered an immune-privileged organ system as the blood–brain barrier (BBB) restricts immune cell traffic into the CNS. However, the BBB can be disrupted in pathological states such as malignant brain tumors, which results in an increased permeability of immune cells into the damaged area [23]. Furthermore, immune cells access the brain through the choroid plexus and structures termed circumventricular organs, which feature fenestrated capillaries without a BBB [24,25]. The CNS primary draining lymph nodes are deep cervical lymph nodes, and mandibular and superficial cervical lymph nodes contribute to antigen sampling in the CNS [26,27,28,29]. Moreover, patients with GBM have tumor regions with both disrupted and intact BBB [30]. These findings support the premise that the BBB cannot be the main factor in the CNS being an immunologically privileged organ system. The predominant immune cells in the brain are microglia and tissue-resident macrophages [31]. The GBM TME comprises diverse cellular components, which include neurons and glial cells (astrocytes, oligodendrocytes, microglia), innate and adaptive immune system cells (monocytes and tumor-associated macrophages [TAMs]), mast cells, neutrophils, and T cells [13,32,33,34]. TAMs are composed of tissue-resident microglia and recruit monocyte-derived macrophages (MDMs) [35], where the microglia-to-MDM ratio in gliomas shifts significantly [13]. Furthermore, gliomas contain an abundance of TAMs and much fewer T cells and NK cells [13,33]. These cell populations render GBM immunologically cold tumors [36]. Moreover, the immunosuppressive GBM TME renders T cells senescent, tolerant, exhausted, and anergic [14]. The GBM immunosuppressive TME is driven by tumor-intrinsic factors and brain tissue responses to tumor antigens, such as overexpression of the indoleamine 2,3-dioxygenase (IDO) enzyme [37] and the oncogene transforming growth factor beta (TGF-β) [38,39]. These findings suggested that microglia, macrophages, and monocytes are the main populations contributing to the immunologically cold feature in the CNS and GBM.

## 3. Immunotherapy against GBM

Immunotherapy mainly aims to selectively eliminate tumor cells based on T cell activation, where the representative immunotherapy involves checkpoint inhibitors, which have been utilized in other solid tumors [7,8]. Checkpoint inhibitors have been studied for treating newly diagnosed and recurrent GBM. The CheckMate-498 study examined the efficacy of nivolumab plus radiation in patients with newly diagnosed O^6^-methylguanine DNA methyltransferase (MGMT)-unmethylated GBM, but did not meet the primary endpoint. Bristol-Myers Squibb announced the CheckMate-548 trial to evaluate patients with the newly diagnosed MGMT-methylated GBM. The CheckMate-143 trial investigated the effect of nivolumab against recurrent GBM as compared to bevacizumab, where the mOS in the nivolumab group was 9.8 months as compared to 10 months in the bevacizumab group [40]. Neoadjuvant use of anti-programmed cell death 1 (PD-1) antibodies revealed that despite the small sample size (35 patients with recurrent GBM), the mOS of patients who received neoadjuvant pembrolizumab was 417 days as compared to 228.5 days in patients who received adjuvant pembrolizumab alone [41]. However, a single-arm study involving the administration of neoadjuvant nivolumab to patients with surgically resectable recurrent GBM reported an mOS of 7.3 months [42]. These studies all analyzed small samples and their results should not be generalized.

High-dimensional proteomics, single-cell transcriptomics, and quantitative multiplex immunofluorescence analysis revealed that neoadjuvant PD-1 blockade resulted in T cell infiltration and the proportion of a progenitor exhausted T cell population, but the majority of infiltrating immune cells were macrophages and monocytes [43]. Interactome analysis demonstrated possible engagement of the cytotoxic T-lymphocyte antigen 4 (CTLA-4) and T-cell immunoreceptor with immunoglobulin and ITIM domains (TIGIT) immune checkpoint pathways [43]. This result indicated that combination therapy with multiple checkpoint inhibitors can potentially overcome GBM. However, this approach might not be adequate to control a highly immunosuppressive TME wherein activated T cells are scarce.

In 2017, the US Food and Drug Administration (FDA) approved CAR-T cell therapies for B cell acute lymphoblastic leukemia (ALL) and large B cell lymphoma [9,44]. Several preclinical and clinical studies investigated GBM treatment, where the specific antigenic targets of CAR-T cells were interleukin-13 receptor alpha 2 (IL-13Rα2), human epidermal growth factor receptor 2 (HER2), erythropoietin-producing hepatocellular carcinoma A2 (EphA2) receptor, and epidermal growth factor receptor variant III (EGFRvIII) [45,46,47,48]. However, GBM CAR-T cell therapy trials did not report similar efficacy to those for hematologic malignancies. This is mainly due to the high degree of intertumoral and intratumoral heterogeneity in GBM, which allows CAR target antigen-non-expressing cancer cells to escape CAR-T attack [49,50]. CAR-T therapy efficacy in solid tumors, and in GBM, is hindered by antigen escape, which leads to antigen-loss tumor cells outgrowth due to the single target antigen [46,51,52]. To overcome the heterogeneous molecular profile, tandem CAR-T cells expressing two antigen-binding domains were designed to resolve antigen escape. Hegde et al. reported the efficacy of combining a HER2 scFv and an IL-13Rα2-binding IL-13 mutant and using CD28 as the costimulatory factor against GBM cells and xenograft GBM [53]. The same group also reported the efficacy of trivalent CAR-T cells that targeted HER2, IL-13Rα2, and EphA2 in GBM cells and patient-derived xenograft models of autologous GBM [54]. Despite the effect of strategies targeting one or a few antigens in solid tumors, the heterogeneous expression of tumor antigens promoted tumor escape following CAR-T cell therapy, particularly in GBM [55]. CAR-T therapies are often accompanied by manageable toxicities, such as cytokine release syndrome (CRS), and are potentially life-threatening [56,57].

## 4. NK Cell and NK Cell-Based Immunotherapy for Cancer

More than 40 years ago, it was determined that NK cells recognize cancer cells in mice and humans without antigen sensitization [58,59,60,61,62]. Recent research focused on the potential of NK cells in cell-based therapies. NK cells are considered an important part of the immune system, where they control microbial infections and tumor progression [58,63]. In patients and animal models, NK cell deficiency or impairment led to recurring virus infections and increased incidence of various types of cancer. In particular, NK cells controlled transplantable tumor growth and metastasis in numerous mouse models via antibody depletion of NK cells [64]. Additionally, NK cells are the founding members of the innate lymphoid cell (ILC) family [65]. In human peripheral blood, bone marrow, and tissues, NK cells can be identified by the expression of neural cell adhesion molecule (NCAM: CD56) and the absence of T cell receptor (TCR) and CD3 [66]. In the bone marrow, human NK cells derive from CD34^+^ hematopoietic progenitors and mature in the lymphoid organs [67,68]. Despite the differentiation from progenitor cells, NK cells persist in peripheral blood [69,70]. Human NK cell turnover in blood occurs over approximately 2 weeks [71].

NK cells can recognize tumor cells based on a balance between stimulatory and inhibitory receptors [stimulatory receptors: DNAX accessory molecule 1 (DNAM1), 2B4 (also known as CD244) and NK group 2D (NKG2D); inhibitory receptors: killer cell immunoglobulin-like receptors (KIRs), TIGIT, killer cell lectin-like receptor subfamily G member 1 (KLRG1), T-cell immunoglobulin mucin family member 3 (TIM3), and programmed death 1 (PD1)] [72,73] (Figure 2). In detail, the main activating receptors are natural cytotoxicity receptors (NCRs: NKp-46/NCR1, NKp44/NCR2, NKp30/NCR3) [74,75,76,77]. B7-H6 and BAG6/BAT3 represent NKp30 ligands [78,79]. NKp44 recognizes a specific human leukocyte antigen (HLA)-DP molecule (HLA-DP401) and PCNA [80,81]. Barrow et al. also reported that platelet-derived growth factor (PDGF)-DD engagement of NKp44 triggered NK cell secretion of interferon (IFN)-γ and tumor necrosis factor alpha (TNF-α), and a distinctive transcriptional signature of PDGF-DD-induced cytokines and the downregulation of tumor cell-cycle genes correlated with NCR2 expression and greater survival in glioblastoma [82]. Gaggero et al. identified the extracellular matrix protein nidogen-1 (NID1) as a ligand of NKp44 [83]. NKp46 binds to the soluble plasma glycoprotein complement factor P/properdin [84]. Garg et al. reported vimentin (a 57-kDA molecule) as a putative NKp46 ligand [85]. The lysis of influenza virus (IV)-infected cells is mediated by the interaction between NKp46, and the IV hemagglutinin (HA) type 1 expressed by the infected cells [86,87,88]. NKG2D is another important NK receptor that transduces activating signals from the transmembrane adaptor protein DAP10 and recognizes UL16-binding proteins (ULBPs) and MHC class-1 related chain (MIC) A/B [89].

Regarding inhibitory receptors, NKG2A–CD94 inhibits NK cell function when bound by HLA-E [90]. However, NKG2C–CD94 heterodimers activate NK cells when bound to HLA-E [91]. In non-HLA-specific inhibitory NK receptors, PD-1, TIGIT, CD96, TIM3, and CD161 function as NK cell activation immune checkpoints, and their ligands are PDL1, Poliovirus receptor (PVR)/PVRL2, galectin-9/high-mobility group protein 1 (HMGB1)/phosphatidylserine (PtdSer), carcinoembryonic antigen-related cell adhesion molecule 1 (CEACAM1), and lectin-like transcript-1 (LLT1), respectively [92]. KLRG1 is another inhibitory receptor expressed by activated NK cells [93]. KLRG1 binds E-cadherin and inhibits human ILC2 function [92]. NK cells express several co-receptors that enhance NK cell triggering activity via NCRs or NKG2D, where the representative co-receptors are 2B4, DNAM-1, and NKp80 [94,95]. Using these receptors, NK cells can recognize whether the adjacent cell (infected or tumor cells) is targeted for killing without prior sensitization. NK cells eliminate cells with diminished or absent MHC-I expression [96]. The MHC-I ligand is a set of KIR inhibitory receptors, which suppress NK cell function and minimize the destruction of healthy self-cells [72,97]. NK cells undergo so-called licensing or education during their development to avoid self-reactivity [98]. NK cells chronically stimulated by self-ligands might become anergic if the inhibitory receptors do not mitigate the stimulation [99,100]. Although the ligation of self-MHC suppresses mature NK cells, the suppression is relieved if MHC is altered or downregulated, which may occur in tumor cells [99,100]. Additionally, the representative classical HLA-F inhibits NK cell function through KIRs [101]. NK cells also have a potent activator, CD16, which recognizes the constant region (Fc) of IgG antibodies and is responsible for antibody-dependent cell-mediated cytotoxicity (ADCC) [102,103].

NK cell functions are regulated by intracellular checkpoint molecules, which are inhibitory signal transduction molecules. Cytokine-inducible SH2-containing protein (CIS) is encoded by cytokine inducible SH2-containing protein (*CISH*) as an IL-15-inducible inhibitor of IL-15 signaling in mouse NK cells [104]. CIS acts as an intracellular checkpoint receptor in tumors with increased IL-15 concentrations [105]. Barsoum et al. reported that reduced nitric oxide levels in prostate cancer cells induced another intracellular checkpoint molecule, hypoxia-inducible factor 1α (HIF1α), which augmented a disintegrin and metalloproteinase domain-containing protein 10 (ADAM10) expression and significantly increased MICA secretion in the extracellular milieu [106]. Moreover, the expression of HIF-1α, a transcriptional factor, promotes multiple signaling and induces immune suppression, including that of NK cells [107,108,109,110].

The hypoxic TME contributes to immune escape in cancer. Casitas B-lineage lymphoma pro-oncogene-b (CBLB) interacts with its specific targets via phosphotyrosine-containing sequence motifs generated on activated protein tyrosine kinases that mediate activating signal transduction [111,112]. In NK cells, CBLB is activated and stabilized through inhibitory receptor signaling and reduces NK cell degranulation and cytotoxicity by targeted degradation of the adaptor protein linker for activation of T cells (LAT) [113,114].

Although NK cells are classified as innate cells, their responses can exhibit the adaptive phenotype of immunological memory or trained immunity under circumstances such as viral infections or stimulation with IL-12, IL-15, and IL-18 cytokines [115,116]. Recently, single-cell RNA sequencing analysis tracked pathogen-specific adaptation within the innate immune system via NK cell memory following human cytomegalovirus infection. NK cell clonal expansion and persistence within the human innate immune system were demonstrated in detail, where these mechanisms evolved independently of antigen-receptor diversification [117].

When NK cells encounter a target cell and are activated, a synapse is formed with the target cell and microtubules transport lytic granules, which converge towards the synapse [118]. Additional signals from the synapse lead to lytic granule polarization; the granules contain the key effectors of cytotoxicity: perforin and granzymes [119,120]. The release of a single granule is sufficient to kill a target tumor cell [119]. Moreover, cytotoxicity is mediated by the death receptors FAS ligand (FasL) and TNF-related apoptosis-inducing ligand (TRAIL) [121]. A marker of this degranulation is cell surface expression of lysosomal-associated membrane protein 1 (LAMP1) [122]. Furthermore, NK cells can secrete cytokines, chemokines, and growth factors, such as IFN-γ, IL-13, TNF, FMS-like tyrosine kinase 3 ligand (FLT3L), CC chemokine ligand 3 (CCL3), CCL4, CCL5, and lymphotactin (XCL1) [72,73,123]. NK cells can activate other immune cells following the secretion of these factors. For example, CCL5 and XCL1 attract DCs and FLT3L increases the number of stimulatory DCs in the TME [11,124]. NK cells are required for the anti-tumor CD8^+^ T cell response by triggering the recruitment of cDC1s and subsequently CD8^+^ T cells [11,12,125,126,127]. The effects of this cascade are highlighted by patient survival across multiple different cancer types, where the gene signatures of cDC1s, NK cells, and CD8^+^ T cells all independently predicted improved survival [11,125,128]. Furthermore, IFN-γ production within the TME upregulates MHC-I [129], which causes tumor evasion from NK cells. However, it also results in activated MHC-I presentation of neoantigens for CD8^+^ T cells. Above all, NK cell-based immunotherapy potentially drives the cancer immunity cycle, indicated by regression and improved patient outcomes.

Unlike T cells, NK cells lack TCRs and do not cause graft-versus-host disease (GVHD) [130,131,132]. NK cell-based immunotherapy presents the possibility of targeting tumors that lack well-defined antigens for specific response and the use of allogeneic products prepared in advance. These facts allow administration in multiple patients without causing GVHD [130,131]. A recent clinical trial demonstrated that ex vivo expanded allogeneic NK cells exhibited enhanced responses against myeloid leukemia. Clinical responses were observed in five of nine evaluable patients, including four complete remissions with low toxicity [133]. Berrien-Elliott et al. reported and summarized the exploration of NK cells as an alternative cell source for allogeneic cell therapies given their inherent ability to recognize cancer, mediate the immune functions of killing and communication, and the fact that they do not induce GVHD, CRS, or immune effector cell-associated neurotoxicity syndrome (ICANS), which indicated low toxicity [134]. Therefore, NK cell-based therapy potentially leads to less toxicity in comparison to CAR-T cell infusions.

## 5. NK Cell-Based Immunotherapy against GBM

GBM is defined as an immunologically cold tumor with immunosuppressive mechanisms [14,135]. In addition to the immunosuppressive TME, GBM is characterized by intratumoral and intertumoral heterogeneity, against which T cell-based therapy (CAR-T therapy and vaccine-based therapy) demonstrates insufficient efficacy. Considering the advantage of NK cell-based immunotherapy, the approach of killing tumor cells is suitable for treating GBM. Specific tumor-associated antigen recognition is not required by NK cells as they recognize tumor cells based on the balance between stimulatory and inhibitory receptors [72,73]. This advantage is used to overcome GBM heterogeneity [21,22], whereas GBM heterogeneity becomes the main issue in CAR-T therapy resistance [55]. In addition to their cytotoxicity, NK cells are required for an optimal anti-tumor CD8^+^ T-cell response by triggering the recruitment of cDC1s and subsequently CD8^+^ T cells [11,12].

NK cells eliminate tumor cells that downregulate their surface expression of MHC-I [136], and the interaction between MHC-I and KIRs suppresses NK cell function and minimizes the destruction of healthy self-cells [72,73]. Given these advantages, several clinical trials investigated NK cell-based immunotherapy efficacy against GBM (Table 1), where NK cells were used as autologous or allogeneic therapy. Allogeneic NK cells are potentially advantageous for immune-suppressed patients because allogeneic NK cells from healthy donor peripheral blood mononuclear cells (PBMCs) have greater ability than exhausted NK cells from the PBMCs of patients with GBM to exert anti-tumor effects against GBM. In this context, cord blood-derived mononuclear cells could also be a promising source of allogeneic NK cell therapy [137].

Another major concern is how the materials used in immunotherapies (checkpoint inhibitors, CAR-T cell therapy, vaccine therapy, NK cell-based therapy) are administered to patients with GBM. In almost every clinical trial, the administration route was mainly intravenous injection [40,46,47,138,139]. In vaccination therapy, intradermal injection is utilized to allow cytotoxic T-lymphocytes (CTLs) to recognize the epitopes associated with HLA class I molecules [140]. Recently, Todo et al. reported the efficacy of G47Δ for residual or recurrent GBM in a phase 2 trial where G47∆ was administered intratumorally via stereotactic surgery [6]. This method was exclusively effective from the direct delivery aspect, which directly affected the immune cells in the immunosuppressive GBM TME.

In the following sections, we focus on adoptive NK cell therapy, CAR-NK cell therapy, checkpoint blockade therapy, and gene-editing NK therapy among the several approaches using NK cell-based therapy (Figure 3).

### 5.1. Adoptive NK Cell Therapy

Adoptive cell therapy using NK cells is categorized into autologous and allogeneic therapy and involves obtaining the patient’s cells and tissues, expanding them ex vivo, and re-infusing them back into the patient. Its advantages include immune system reaction, biocompatibility, and no disease transmission risk [141], while the disadvantages are the complicated set-up, the requirement for well-trained researchers, and the high treatment cost. Another issue is the inhibitory response that results from the recognition of self MHC molecules [142,143].

NK cells comprise a minority of blood lymphocytes (approximately 10%), and their extraction from PBMCs is difficult, particularly in obtaining sufficient numbers to utilize in cancer treatment in vivo. IL-2 induces NK cell proliferation, but this stimulation is not consistent or sustained [144,145]. Proliferative responses in NK cells can be rapid and sustained when they are co-cultured with stimulatory cells, such as the Wilms tumor-derived cell line, autologous PBMCs, K562 cells, [146] and Epstein–Barr virus-transformed lymphoblastoid cells [147,148,149,150,151]. Given the difficulty of their complicated PBMC-derived ex vivo expansion and culture, there have been few clinical trials based on autologous NK cells. Ishikawa et al. reported the expansion of NK cell-rich effector cells ex vivo from the PBMCs of patients with recurrent malignant glioma [152]. They concluded that NK cell therapy was safe and partially effective in patients with recurrent malignant gliomas [152]. In that study, NK cells were cultured with a human feeder cell line (HFWT) as feeder cells [152].

Lymphokine-activated killer (LAK) cells constitute another type of autologous therapy. A phase 2 trial involving patients with recurrent glioma reported a higher survival rate in the patients who received LAK cells and IL-2 to the CNS [153,154]. We reported genuine induced NK cells (GiNK), which are highly purified human NK cells derived from PBMCs using a feeder-free method and that exhibited high NK activity for GBM cells [155,156], where our expansion method rapidly yielded a large number of purified NK cells. We also reported a robust large-scale feeder-free expansion system for highly purified human NK cells using a defined cytokine cocktail (IL-12 and IL-18) and anti-NK cell activating receptor antibodies (anti-NKp46 and/or anti-CD16 antibody) [157]. In that report, all NK cells tested markedly inhibited T98G cell growth. These methods could provide novel insights into NK cell-based immunotherapy. In a murine model of xenograft GBM, systemically injected IL-2/HSP70-treated NK cells reduced the number of GBM cells [158].

Another approach is the differentiation of NK cells from stem cells, such as induced pluripotent stem cells (iPSCs), human embryonic stem cells (hESCs), hematopoietic stem cells, and cord blood cells [159,160,161]. Recently, the FDA approved the use of allogeneic NK cells differentiated from hematopoietic stem cells from human placenta for GBM treatment. Established human cell lines, such as the NK-92 cell line, isolated from a 50-year-old male patient with rapidly progressing non-Hodgkin lymphoma [162], were utilized in clinical and preclinical trials.

### 5.2. CAR-NK Cell Therapy

While immune cell genetic engineering mainly focused on T cells, NK cells are potent effectors of antitumor immune responses. Several studies successfully used NK cells engineered to express activating CARs [163], which are engineered molecules that consist of four main components: a single-chain variable fragment, the hinge or space, a transmembrane domain, and one or more intracellular signaling domains [164]. The target molecules of CAR-NK cells using NK-92 cells were CD19, CD20, HER2, EPCAM, and CD138 [163,165,166,167,168]. When tested in xenograft models in vivo, CAR-NK cell therapies using primary human NK cells targeting CD19, HER2, and mesothelin yielded increased responses to tumor cells in vitro and suppressed tumor growth [169,170,171,172]. CARs exhibited significantly stronger NK cell cytotoxicity than ADCC mediated by antibodies against the same target [173]. Unlike T cells, NK cells cannot produce autocrine growth factors; therefore, CAR-NK cells exert limited off-target adverse effects [174]. Even if the targeted antigens were lost on tumor cells, CAR-NK cells can nevertheless be stimulated by their activating receptors. Another advantage of CAR-NK therapy is the use of allogeneic cells without causing GVHD [175,176]. Compared to autologous therapy, allogeneic therapy demonstrated greater ability to exert anti-tumor effects against GBM.

Most studies on CAR engineered effector cells in GBM focused on autologous CAR-T cells [46,47,153,177]. IL-13Rα2 was first reported as a therapeutic target in GBM [178] and is overexpressed in >50% of high-grade gliomas [179]. A phase 1 clinical trial of autologous CAR-T cells targeting IL-13Rα2 against recurrent GBM reported GBM regression [51]. *EGFR* gene amplification and EGFR protein overexpression is present in 40–60% of GBM [180]. A pilot trial where recurrent EGFRvIII-positive GBM was treated with EGFRvIII-specific third-generation CAR-T cells with IL-2 did not report a clinically meaningful effect in patients with GBM [181]. CAR-NK therapy focused on EGFRvIII and HER2 as its target molecules in GBM treatment, where intravenous injection into mice carrying subcutaneous EGFRvIII-positive GBM xenografts inhibited tumor growth and extended OS [182]. Additionally, co-expressing the chemokine receptor CXCR4 in the CAR-NK cells for improved tumor-homing enhanced the anti-tumor effect. Engineered KHYG-1 NK cells with EGFRvIII-specific CAR enhanced cytotoxicity against EGFRvIII-expressing GBM cells in vitro and induced a pseudoprogression-like feature in subcutaneous GBM-like tumors in vivo [183,184]. Repeat injection of NK-92 cells bearing EGFRvIII-specific CAR into EGFRvIII-positive GBM xenografts inhibited tumor growth and improved symptom-free survival [185]. However, treatment against mixed tumors consisting of EGFR and EGFRvIII double-positive and EGFR-positive EGFRvIII-negative GBM cells yielded less survival benefit and tumor growth inhibition [46]. To resolve this issue, Han et al. reported a cell-binding domain of second-generation CAR with CD28 and CD 3ζ signaling domains and a scFv fragment of the EGFR-specific antibody 528 [186]. In that study, NK-92 and NKL cells expressing this CAR exhibited enhanced cytotoxicity and IFN-γ secretion against GBM cell lines or patient-derived GBM stem cells expressing EGF or EGFRvIII. In another study, a scFv fragment derived from the clinically applied antibody cetuximab was used as a cell-binding domain for a dual-specificity second-generation CAR with CS28 and CD 3ζ signaling domains [185]. In that report, local treatment with dual-specificity NK cells was superior to treatment with the corresponding monospecific CAR-NK cells in immunodeficient mice carrying intracranial GBM xenografts expressing EGFR, EGFRvIII, or both.

In addition to EGFRvIII and EGFR, HER2 has been targeted with CAR-engineered NK-92 cells. In preclinical studies, Schönfeld et al. followed good manufacturing practice (GMP)-compliant procedures and generated a stable monoclonal cell line expressing a humanized CAR based on the ErbB2-specific antibody FRP5 harboring CD28 and CD 3ζ signaling domains (CAR 5.28.z) [168]. The cells demonstrated high, selective cytotoxicity against ErbB2-positive target cells from different solid tumors, including GBM cell lines [168,187]. In orthotopic GBM xenograft models, repeated stereotactic injection of the cells into the tumor effectively inhibited tumor growth and prolonged OS [187].

The phase 1 clinical trial CAR2BRAIN (NCT03383978, ClinicalTrials.gov) was an investigator-initiated, prospective, open-label study performed at the Center for Neurology and Neurosurgery at the University Hospital Frankfurt, Germany. The study involved patients with recurrent or refractory ErbB2-positive GBM with scheduled relapse surgery. The main purpose of the study was to evaluate the safety and tolerability of ErbB2-specific NK-92/5.28.x CAR-NK cells [168,188]. NK-92/5.28.z cells were grown to clinically applicable doses of 5 × 10^8^ cells/L in a 5-day batch culture without loss of viability and potency, then injected into the resection cavity wall during relapse surgery. Doses of 1 × 10^7^, 3 × 10^7^, and 1 × 10^8^ in a total 2-mL injection volume were examined. The NK-92/5.28.z maintenance culture enabled expansion of the therapeutic dose to up to 5 × 10^9^ cells in 10 L within 5 days. This was the only CAR-NK clinical trial that analyzed CAR-NK therapy in patients with GBM.

Although CAR-T therapy was investigated in several solid tumors, including GBM, clinical studies that focused exclusively on CAR-NK therapy were limited. Despite the multiple advantages compared to CAR-T therapy, one remaining challenge is the transduction efficacy of CAR receptor gene insertion. Gene modification of primary NK cells using viral or non-viral vectors is challenging due to robust foreign DNA- and RNA-sensing mechanisms that limit the efficiency of these gene delivery methods [189]. Although lentiviral and retroviral transduction of primary NK cells yielded improvements, CAR expression remained low. Naeimi Kararoudi et al. reported mRNA-based gene delivery, but this approach enabled only transient transgene expression [190]. The authors also reported CAR transduction into primary NK cells using CRISPR and adeno-associated viruses (AAV), which demonstrated high transduction efficiency [191]. Above all, CAR-NK therapy would overcome the difficulty of CAR receptor gene insertion transduction methods. Recently, Shimasaki reported the gene transduction of K562 cells expressing 4-1BBL and membrane-bound IL-15 (K562-mb15-41BBL), where NK cells were stimulated with a retrovirus vector [192]. The method was complicated but nevertheless demonstrated high transduction efficacy in human NK cells. Baboon envelope pseudotyped lentivirus vector also enhanced the efficacy of gene transduction into human NK cells [193].

### 5.3. Checkpoint Blockade Therapy

The checkpoint receptor blockade exerts a substantial effect on malignant tumors. However, few studies reported the effectiveness of checkpoint inhibitors in GBM treatment. Considering the poor population of NK cells in GBM [13,33,34,50], adoptive NK cell therapy with checkpoint inhibitors to attack the immunosuppressive GBM TME is a promising approach.

Shevtsov et al. reported that combination therapy of NK cells activated by IL-2 and TDK derived from HSP70 and anti-PD1 antibody exhibited stronger cytotoxicity effects against intracranial GBM GL261 cells in vitro [194]. They also demonstrated that combination therapy of NK cells and anti-PD1 antibodies reduced tumor volume and improved OS in an immunodeficient mouse GL261 xenograft model. We also investigated the combination therapy of our established highly activated and purified NK cells and anti-PD1 antibodies against GBM cell lines in vitro and in vivo [195]. In that report, PD-1–PD-L1 axis blockade did not enhance the anti-tumor activity of our established primary NK cells in vitro and in vivo. The result indicated that the PD-1–PD-L1 axis did not affect our established NK cells, but the additional effect of PD-1 blockade in the clinical setting is controversial.

In addition to PD-1 blockade therapy, other checkpoint inhibitors have been investigated. André et al. reported that blocking the inhibitory NKG2A receptor enhanced tumor immunity by promoting both NK and CD8^+^ cell effector functions in mice and humans [196]. The humanized anti-NKG2A antibody monalizumab enhanced NK cell activity against various tumor cells [196]. Romagné et al. demonstrated that 1-7F9 conferred specific, stable blockade of KIR, which boosted NK-mediated killing of HLA-matched acute myeloid leukemia (AML) blasts in vitro and in vivo [197]. In GBM, autologous NK cells and T cells suppressed the tumor by expressing MHC-I molecules; checkpoint inhibitor blockades would be advantageous in adoptive NK cell therapy.

### 5.4. Genome-Editing NK Cell Therapy

This section summarizes genome-editing NK cell therapy, other than CAR-NK therapy. The combination of adoptive NK cell therapy and checkpoint inhibitors for treating GBM is attractive, but checkpoint inhibitor efficacy is dependent on the timing of administration and delivery efficacy, especially in CNS tumors. However, genome editing NK cells, which knocks out the immunosuppressive receptor genes, can sustainably prevent inhibitory receptor expression.

The CRISPR/Cas9 genetic-editing system is widely utilized to genome-edit T cells to disrupt inhibitory genes such as *PD1* and *CTLA4* [198,199,200,201,202]. Unlike T cell CRISPR/Cas9 genome-editing, NK cell CRISPR-editing has been challenging. The issue in NK cell genome editing is low viral transduction efficiency, where Cas9/RNP electroporation yielded new insights [191,203,204,205,206]. Rautela et al. first reported on CRISPR/Cas9 genome-editing of primary human NK cells targeting CISH and a representative NKp46 NK receptor [206]. Two groups reported CD38 knockout NK cells using CRISPR/Cas9 via electroporation [205,207]. They demonstrated that the CD38 knockout NK cells exhibited a more prominent metabolic profile, which increased ADCC-mediated killing of CD38^+^ multiple myeloma cell lines and patient-derived samples [205,207]. Zhu et al. reported that *CISH* knockout iPSC-derived NK cells demonstrated significantly increased in vivo persistence and inhibition of tumor progression in a leukemia xenograft model [208]. Pomeroy et al. knocked out *PD1* in NK cells by electroporating *CAS9* mRNA and genomic RNA (gRNA), where the *PD1* knockout NK cells demonstrated significantly enhanced cytotoxicity and cytokine secretion in vitro and in vivo with reduced tumor volume and prolonged OS [209]. Table 2 summarizes the aforementioned CRISPR/Cas9-mediated NK cell-based therapies according to checkpoint receptors and intracellular checkpoint molecules.

To utilize this approach for GBM treatment, we first reported the establishment of *TIM3* knockout primary NK cells and enhanced cytotoxicity against GBM cell lines [155]. In our setting, 10^6^ order of NK cells per shot underwent electroporation and we were able to obtain 4.7 × 10^8^ genetically modified NK cells from 16 mL blood in 2 weeks, which was sufficient for clinical application. In comparison, a previous report documented an electroporation setting of 10^5^ order of NK cells per shot and substantially lower expansion efficacy [206]. Huang et al. reported CRISPR/Cas9 knockout of multiple genes (*TIGIT* and *CD96* double knockout and *TIGIT*, *CD96*, and *CD226* triple knockout) in primary NK cells [210]. The cytotoxicity of the genome-edited NK cells against the U251MG GBM cell line was investigated, where the multiple-knockout NK cells did not exhibit crucial activation [210].

Although CRISPR/Cas9 genome-editing in primary NK cells lags behind that of T cells, CRISPR/Cas9 technology validation is evolving and this approach could be promising for GBM treatment.

**Table 2 ijms-24-02111-t002:** CRISPR/Cas9 electroporation genome-editing NK cell-based immunotherapy against all cancers, including GBM.

Target	Source of NK Cells	Cancer Types	References
Intracellular receptors			
CISH	Cord blood-derived CAR-NK cells	Lymphoma	[211]
	PBMCs isolated from healthy donor	Lymphoma	[206]
	human iPSCs	AML, CML, ovary cancer	[208]
CBLB	NK cells derived from placenta-derived CD34^+^ hematopoietic stem cells	AML	[212]
PRDM1	PBMCs isolated from healthy donor	None	[213]
Extracellular receptors			
CD38	PBMCs isolated from healthy donor	MM	[205]
	PBMCs isolated from healthy donor	AML	[207]
PD1	PBMCs isolated from healthy donor	AML, lymphoma, prostate cancer	[209]
	PBMCs isolated from healthy donor	AML, breast cancer, GBM	[210]
NKG2A	PBMCs isolated from healthy donor	AML	[214]
	PBMCs isolated from healthy donor	AML, breast cancer, GBM	[210]
NCR1/NKp46	PBMCs isolated from healthy donor	Lymphoma	[206]
TIM3	PBMCs isolated from healthy donor	GBM	[155]
TIGIT, CD96, CD226, SIGLEC-7	PBMCs isolated from healthy donor	AML, breast cancer, GBM	[210]
TGF β receptor	PBMCs isolated from healthy donor	GBM	[215]
ADAM17	PBMCs isolated from healthy donor	AML, lymphoma, prostate cancer	[209]

ADAM17: A disintegrin and metalloproteinase domain-containing protein-17; AML: acute myeloid leukemia; CAR-NK: chimeric antigen receptor natural killer; CBCL: Casitas B-lineage lymphoma pro-oncogene-b; CISH: cytokine-inducible SH2-containing protein; CML: chronic myeloid leukemia; GBM: glioblastoma; iPSCs: induced pluripotent stem cells; MM: multiple myeloma; NCR1: natural cytotoxicity triggering receptor 1; PD1: programmed death 1; PRDM1: PR/SET domain 1; TGF: transforming growth factor; TIGIT: T cell immunoreceptor with Ig and ITIM domains; TIM3: T-cell immunoglobulin and mucin domain-containing 3.

## 6. Future Perspectives

While several immunological approaches for treating GBM have long been investigated, the challenges to overcoming GBM persist. These immunotherapies focus on activating the T-cell immune response, which involves the specific antigen recognition process of T cells. Accordingly, T cell-based immunotherapies are limited due to the high heterogeneity of glioma cells. Consequently, patients with GBM cannot acquire an ideal response to immunotherapy, which results in specific antigen loss. NK cells can resolve this issue of recognizing tumor cells by balancing multiple activating and inhibitory receptors. Furthermore, NK cell activation would activate adaptive immunity and can mobilize a variety of immune cells to exert anti-tumor effects. Recently, it became possible to produce activated human primary NK cells in vitro with high purity and expansion efficiency, and these NK cells have been actively applied in clinical research. Despite the exclusive immunosuppressive condition of the GBM TME, which may render NK cells dysfunctional or tolerogenic, several techniques, including checkpoint inhibitors or genome editing, are being investigated to address the dysfunctional status of NK cells. Overall, NK cell-based immunotherapy, in the field of GBM treatment, is promising, and could achieve a breakthrough in GBM immunotherapy.

## 7. Conclusions

The standard GBM treatment is insufficient to rescue patients; therefore, novel strategies should be investigated. We highlighted NK cell-based immunotherapy against GBM and emphasized the potential of NK cell activity in eliciting tumor killing and tumor inflammation. The standard GBM treatment is insufficient to rescue patients; therefore, novel strategies should be investigated. Considering GBM intertumoral and intratumoral heterogeneity and the immunosuppressive TME, NK cell-based immunotherapy is a more suitable approach for treating GBM. Additionally, NK cells subsequently induce the anti-tumor CD8^+^ T cell response and other immune cells. NK cell-based immunotherapy can change the GBM TME from an immunologically “cold” tumor into a “hot” tumor. Nevertheless, several challenges persist in efforts to overcome GBM.

## Figures and Tables

**Figure 1 ijms-24-02111-f001:**
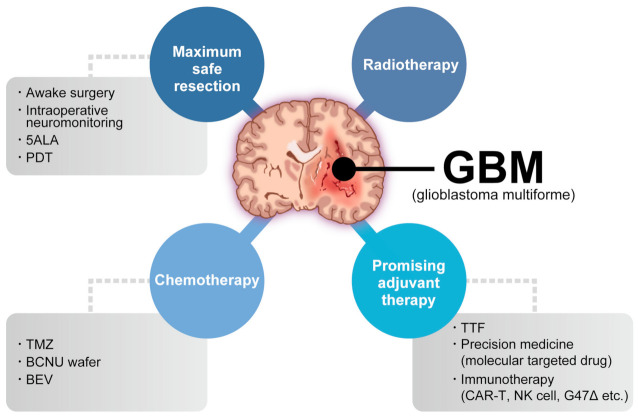
Multimodality treatment against GBM. While the standard therapy consists of maximum safety resection, adjuvant radiotherapy, and chemotherapy with temozolomide, which is collectively termed the Stupp regimen, several promising adjuvant therapies have been used for treating GBM. 5ALA: aminolevulinic acid, PDT: photodynamic therapy, TTF: tumor-treating fields, TMZ: temozolomide, BCNU: carmustine, BEV: bevacizumab.

**Figure 2 ijms-24-02111-f002:**
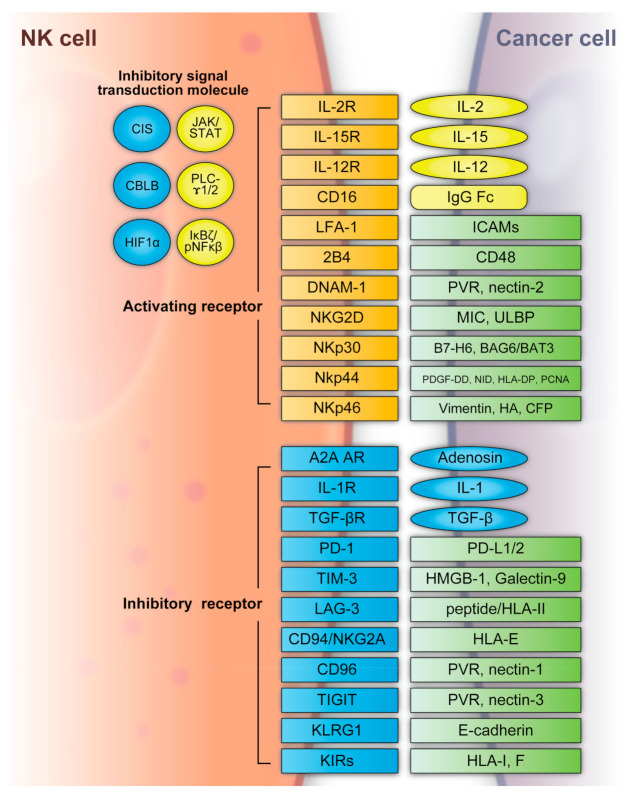
NK cell activating and inhibitory receptors. NK cells recognize tumor cells based on a balance between the stimulatory and inhibitory receptors above. A2A AR: A2A adenosine receptor, BAG6/BAT3: BCL2-associated athanogene cochaperone 6, CBLB: Casitas B-lineage lymphoma pro-oncogene-b, CIS: cytokine-inducible SH2-containing protein, DNAM1: DNAX accessory molecule 1, HA: IV hemagglutinin, HIF1α: hypoxia-inducible factor 1α, HLA: human leukocyte antigen, HLA-DP: human leukocyte antigen DP molecule, HMGB1: high-mobility group protein 1, ICAMs: intracellular adhesion molecules, IgG Fc: constant region of immunoglobulin, IL: interleukin, KIR: killer cell immunoglobulin-like receptors, JAK/STAT: Janus kinase/signal transducer and activator of transcription, KLRG: killer cell lectin-like receptor subfamily G member 1, LAG3: lymphocyte-activation gene 3, LFA-1: lymphocyte function-associated antigen-1, MIC: major histocompatibility complex class 1-related chain, NF-κB: nuclear factor kappa B, NID: nidogen, NKG2D: NK group 2D, NKp30: natural killer cell p30-related protein, NKp44: natural killer cell p44-related protein, NKp46: natural killer cell p46-related protein, PCNA: proliferating cell nuclear antigen, PD-1: programmed cell death 1, PDGF-DD: platelet-derived growth factor, PD-L1/2: programmed cell death ligand 1/2, PLC-γ1/2: phospholipase C γ1/2, PVR: poliovirus receptor, TGFβ: transforming growth factor β, TIGIT: T-cell immunoreceptor with immunoglobulin and ITIM domains, TIM3: T-cell immunoglobulin mucin family member 3, ULBP: UL16-binding protein.

**Figure 3 ijms-24-02111-f003:**
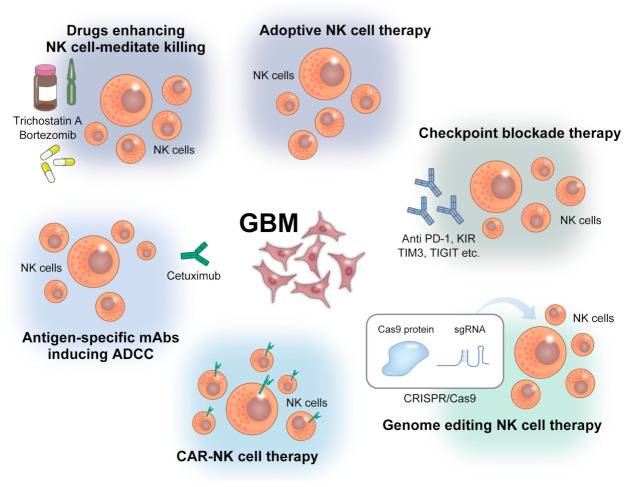
Strategies of NK cell-based immunotherapy against GBM. ADCC: Antibody-dependent cellular cytotoxicity, PD-1: programmed cell death 1, KIR: killer cell immunoglobulin-like receptors, TIGIT: T-cell immunoreceptor with immunoglobulin and ITIM domains, TIM3: T-cell immunoglobulin mucin family member 3.

**Table 1 ijms-24-02111-t001:** Active clinical trials of NK cell-based immunotherapies against glioma.

Trial	Phase	Administration Route	NK Cell Source	Agent	Tumor
CAR NK cell therapy					
NCT03383978	I	Intracranial injection	NK-92 cells	Anti-MUC1 CAR NK cells	MUC1+ glioma
NCT02839954	I/II	Venous infusion	Autologous NK cells	NK-92/5.28.z cells	Recurrent or refractory HER2^+^ GBM
Adoptive NK cell therapy					
NCT00909558	I	Venous infusion	Autologous NK cells	Autologous NK cells	Glioma
NCT04254419	I	Intra-tumoral injection	Autologous NK cells	Autologous NK cells	Malignant glioma
NCT01875601	I	>Venous infusion	>Autologous NK cells	>Autologous NK cells with rhIL-15	>Advanced solid tumors
NCT02100891	II	Venous infusion	Autologous NK cells	Autologous NK cells with HLA haploidentical HCT	High-grade glioma
LAK cell therapy					
NCT00814593	II	Intra-tumoral injection	Autologous PBMCs	LAK cells	Primary GBM
NCT00003067	II	Intra-tumoral injection	Autologous PBMCs	LAK cells with aldesleukin	Primary, recurrent, or refractory malignant gliomas
NCT00331526	II	Intra-tumoral injection	Autologous PBMCs	LAK cells with aldesleukin	Primary GBM
NCT00005813	I	Intra-tumoral injection	Autologous PBMCs	LAK cells with MDX477 bispecific antibody	EGFR-expressing GBM

EGFR: Epidermal growth factor receptor; GBM: glioblastoma; HCT: hematopoietic cell transplantation; HER2: human epidermal growth factor type 2; LAK: lymphokine-activated killer; MUC1: mucin 1; NK, natural killer; PBMC: peripheral blood mononuclear cells; rhIL-15: recombinant human IL-15.

## Data Availability

Not applicable.

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
