# Peer review of "Natural Killer Cell-Based Immunotherapy against Glioblastoma"

_ijms, 2023, doi:10.3390/ijms24032111_

Round 1

Reviewer 1 Report

Comments and Suggestions for Authors

The manuscript presented by Takayuki Morimoto et al descripted that Natural killer (NK) cell-based immunotherapy features the new aspect of recognizing GBM via mechanisms that differ from that of T cell-based immunotherapy. Glioblastomas remain the most lethal primary brain tumors. NK cell-based therapy is a promising immunotherapeutic strategy in the treatment of glioblastoma. This manuscript is very interesting topic and well organized. Below are some comments to help strengthen the manuscript.

1.     Although the immune system can detect and eliminate cancer cells, the microenvironment of the glioblastoma could suppress through diverse mechanisms such as the secretion of a large number of substances that interact with NK cells blocking their action. It is better to add the mechanisms of immunosuppression in GB microenvironment.

2.     Figure Legends are way too vague.

Reviewer 2 Report

Comments and Suggestions for Authors

In this article, Morimoto et al. review the role of NK cells as a novel immunotherapy for glioblastomas (GBMs). The review is well articulated and covers important topics, including existing immunotherapies and the potential of NK-based therapies. This is a timely review as solid tumors like glioblastomas are still a matter of concern because of the lack of effective treatments and the resultant lower survival rate. The authors need to address the following points to get the article accepted:

  1. Section 5 covers many important topics. Adding 1-2 associated figures explaining different therapies would improve the understanding.
  2. A section on caveats (or potential caveats) of NK-based therapies is warranted to keep the article balanced.

Reviewer 3 Report

Comments and Suggestions for Authors

The Manuscript by Morimoto et al. reviews the current progress of GBM immunotherapy based Natural Killer T Cells (NKTs). Although the content is up-to-date and informative, I have some concerns regarding general structure of this Manuscript, citation errors, irrelevant data, and no “future perspectives” section which would certainly bring more attention from the scientific community, especially in such a hot topic.

1. There are some repetitions across the Manuscript (f.ex., lines 41-42 and 102-103). I suggest the Manuscript is re-read by one Author to better unify the text across chapters and avoid unnecessary repetitions.

2. I failed to find how Reference [6] Kimura et al. 2022 (line 38) reflects the G47Δ cancer study. Shouldn’t this be Todo et a. 2022 (DOI: https://doi.org/10.1038/s41591-022-01897-x), see line 316. Also, shouldn’t Reference [5] Stupp et al. 2017 (line 35) concern more the data given in sentence across lines 29-31? Again, I suggest the Manuscript is re-read by one Author to verify the remaining citations across the text, as there may be more oversights.

3. The Authors only briefly referred to single-cell transcriptomics (line 119) which, in the context of GBM immunotherapy, seems like an oversight. Please refer to, f.ex., Verhaak et al. 2010 (DOI: 10.1016/j.ccr.2009.12.020) and Wang et a. 2017 (DOI: 10.1016/j.ccell.2017.06.003). I suggest the Authors to briefly elaborate on NKTs’ potential against each of the GBM subtypes (Proneural, Classical, and Mesenchymal).

4. Apart from References [209], [154] and [214], Table 2 is largely irrelevant to the topic of brain cancers, f.ex., was CIS-expressing gene (CISH) even studied in the context of GBM? Unless the Authors can elaborate on why the remaining targets are relevant to GBM, I doubt Table 2 in its current form would be of interest to the Reader. Especially as References [209], [154] and [214] are already appropriately described in the body of the text. Alternatively, the Authors may consider adjusting the title to attract a broader audience, or limit Table 2 to just the examples relevant to brain cancers.

5. The Review lacks a speculative “future challenges”/”future perspectives” kind-of-chapter with Authors’ own critical thoughts. Due to the enormous clinical relevance of NKT-based GBM therapy, maybe the Authors could expand the article’s ending and elaborate on how would NKT-based immunotherapy potentially fit into the current GBM treatment regimen?

6. The central nervous system (CNS) is not an organ per se, but rather a group of organs, or an organ system (line 76; lines 85-86), please correct.

7. Figure 1: According to Todo et al. 2022 (DOI: https://doi.org/10.1038/s41591-022-01897-x) it should be “G47Δ”, not “GΔ47”.

8. Regarding Table 2, since electroporation is consistent throughout the given references, the column “Method” seems redundant, and could be just mentioned once in the table caption for simplicity.

9. Table 1 should be horizontal so that the words don’t overlap.

10. The abbreviation “HFTW” (line 339) is not defined.

Round 2

Reviewer 3 Report

Comments and Suggestions for Authors

The Author's managed to respond to all my comments adequatly. I believe the Manuscript demonstrates a substantial contribution to the field and fully recommend its publication in the current form.

Regarding Comments in Chapter 6:

[E1] Refering to limited efficacy is fine.

[E2] Substitution is ok.